# A Novel Predictive Model of Pathological Lymph Node Metastasis Constructed with Preoperative Independent Predictors in Patients with Renal Cell Carcinoma

**DOI:** 10.3390/jcm12020441

**Published:** 2023-01-05

**Authors:** Jian-Xuan Sun, Chen-Qian Liu, Zong-Biao Zhang, Qi-Dong Xia, Jin-Zhou Xu, Ye An, Meng-Yao Xu, Xing-Yu Zhong, Na Zeng, Si-Yang Ma, Hao-Dong He, Wei Guan, Shao-Gang Wang

**Affiliations:** Department and Institute of Urology, Tongji Hospital, Tongji Medical College, Huazhong University of Science and Technology, Wuhan 430030, China

**Keywords:** renal cell carcinoma, lymph node metastasis, nomogram, lymph node dissection, preoperative predictive model

## Abstract

**Introduction:** Renal cell carcinoma (RCC) is one of the most common urinary tumors. The risk of metastasis for patients with RCC is about 1/3, among which 30–40% have lymph node metastasis, and the existence of lymph node metastasis will greatly reduce the survival rate of patients. However, the necessity of lymph node dissection is still controversial at present. Therefore, a new predictive model is urgently needed to judge the risk of lymph node metastasis and guide clinical decision making before operation. **Method:** We retrospectively collected the data of 189 patients who underwent retroperitoneal lymph node dissection or enlarged lymph node resection due to suspected lymph node metastasis or enlarged lymph nodes found during an operation in Tongji Hospital from January 2016 to October 2021. Univariate and multivariate logistic regression and least absolute shrinkage and selection operator (lasso) regression analyses were used to identify preoperative predictors of pathological lymph node positivity. A nomogram was established to predict the probability of lymph node metastasis in patients with RCC before surgery according to the above independent predictors, and its efficacy was evaluated with a calibration curve and a DCA analysis. **Result:** Among the 189 patients, 54 (28.60%) were pN1 patients, and 135 (71.40%) were pN0 patients. Three independent impact factors were, finally, identified, which were the following: age (OR = 0.3769, 95% CI = 0.1864–0.7622, *p* < 0.01), lymph node size according to pre-operative imaging (10–20 mm: OR = 15.0040, 95% CI = 1.5666–143.7000, *p* < 0.05; >20 mm: OR = 4.4013, 95% CI = 1.4892–7.3134, *p* < 0.01) and clinical T stage (cT1–2 vs. cT3–4) (OR = 3.1641, 95% CI = 1.0336–9.6860, *p* < 0.05). The calibration curve and DCA (Decision Curve Analysis) showed the nomogram of this predictive model had good fitting. **Conclusions:** Low age, large lymph node size in pre-operative imaging and high clinical T stage can be used as independent predictive factors of pathological lymph node metastasis in patients with RCC. Our predictive nomogram using these factors exhibited excellent discrimination and calibration.

## 1. Introduction

Renal cell carcinoma (RCC) is a kind of cancer derived from renal epithelial cells, accounting for more than 90% of all renal malignancies, which is one of the most common cancer types and accounts for 2–3% of systemic malignancies. Metastasis will occur in about 1/3 of all patients with RCC, and 30–40% of those with metastasis will have lymph node metastasis [1,2]. The incidence of lymph node metastasis in RCC significantly increases with the increase of tumor T stage. The probability of lymph node metastasis in pT1-pT3 stage was 1.1%, 4.55% and 12.3%, respectively [3]. Once lymph node metastasis occurs, the 5-year survival rate ranges from 5% to 38%, which is an independent risk factor for poor prognosis [4]. Lymph node metastasis would significantly reduce the survival time of patients. Therefore, an accurate evaluation of lymph node metastasis before operation has an important guiding significance for clinical decision making. However, there is still controversy about whether to perform lymph node dissection during RCC surgery. The role of lymph node dissection in the prognosis of RCC remains not clear. The main purpose of lymph node dissection is to help clarify the clinical stage. Previous studies have shown that lymph node dissection has little effect on the overall survival and recurrence-free survival of RCC patients [5]. Moreover, expanding the scope of surgery can also increase the risk of surgery, damage the immunity of patients and increase the incidence of perioperative complications [4]. Therefore, how to accurately identify patients who need lymph node dissection or biopsy is very important. Foreign reports suggest that tumor diameter, clinical T stage, pathological type, sarcomatoid change and pathological necrosis are closely related to lymph node involvement [5,6]. However, some factors in the above predictive model cannot be obtained before operation, which limits its clinical application. Therefore, we need to build a novel predictive model to predict the risk of lymph node metastasis and assist clinical decision making before operation. In this study, we retrospectively collected and analyzed the clinical data of patients in Tongji Hospital, Tongji Medical College, Huazhong University of Science and Technology from January 2016 to October 2021 and successfully found the independent factors connected with lymph node involvement. A nomogram was finally established to predict the probability of lymph node metastasis in patients with RCC before surgery according to the above independent predictors, and its efficacy was evaluated with a calibration curve and DCA analysis.

## 2. Materials and Methods

Initially, a total of 1426 patients who underwent radical nephrectomy or partial nephrectomy because of renal cell carcinoma in our hospital were retrospectively collected. Subsequently, 189 patients who underwent retroperitoneal lymph node dissection or enlarged lymph node resection due to suspected lymph node metastasis or enlarged lymph nodes found during operation were selected for further analyses. The basic demographic data, preoperative routine examination and examination data and postoperative pathological results of the patients were collected. All pathological information was collected by professional pathologists. Among those patients, 121 were males and 68 were females, with a median age of 54, and there were 58 cases with an Eastern Cooperative Oncology Group (ECOG) score ≥ 1.

Only lumbar and abdominal pain or hematuria were defined as local symptoms; the simultaneous occurrence of low back pain and hematuria was defined as systemic symptoms. The other systemic symptoms included fever, fatigue, multiple bone pain and significant weight loss. A discovery from physical examination or accidental examination was defined as asymptomatic. Age-adjusted Charlson comorbidity index (aCCI)—a widely used scoring system for complications that quantifies the complications based on the number and severity of patients’ diseases and can be used to predict the risk of disease death—was used to divide hypertension and diabetes into 3 grades, namely mild comorbidity (0–1), moderate comorbidity (2–3) and severe comorbidity (≥4) [7].

Statistical Products and Services Solutions (SPSS) version 25.0 and R software 4.1.1 were performed to analyze all statistics. Youden index was used to determine the cut-off value. The difference among continuous variables with normal distribution (expressed as mean ± SD) was detected with Student *t*-test. Continuous variables of skewed distribution were exhibited as median (interquartile range [IQR]) and compared with Mann–Whitney U test, and the difference (expressed in proportion) among categorical variables groups was detected via Chi square test or Fisher’s exact test. Kruskal–Wallis H rank sum test was performed to compare ranking order variables (expressed in proportion). We used the univariate logistic regression and the least absolute shrinkage and selection operator (lasso) regression method to screen out the candidate risk factors and then carried out multivariate logistic regression analysis. We selected the candidate object with non-zero coefficient to establish the lasso model [8]. Multivariate logistic regression was used to determine the independent factors of lymph node metastasis. Then we converted each regression coefficient in multivariable logistic regression into a scale of 0–100 points in proportion and established a predictive nomogram [9]. Futhermore, we applied calibration curve, receiver operating characteristic (ROC) curve and decision curve analysis (DCA) to evaluate the predictive efficiency and clinical practicability of this model [10]. When *p* < 0.05, the difference was considered statistically significant. At the same time, we also calculated the C index of different prediction indicators. Finally, we plotted a decision tree according to the independent factors.

## 3. Result

As shown in Table 1, among the 189 patients collected, 54 (28.60%) were pN1 patients, and 135 (71.40%) were pN0 patients. A total of 271 lymph nodes were resected in 54 patients with positive pathological lymph nodes, including 163 positive lymph nodes. Among the pN1 patients, 19 (35.2%) patients were of papillary renal cell carcinoma, 14 (25.9%) patients were of clear cell carcinoma, 1 (1.9%) patient was of renal medullary carcinoma, 8 (14.8%) patients were of MiT family translocation renal cell carcinoma, 3 (5.6%) patients were of collecting duct carcinoma and 9 patients belonged to unclassified types. A total of 532 lymph nodes were removed in 135 patients with negative pathological lymph nodes. Among those patients, there were 6 (4.4%) patients of papillary renal cell carcinoma, 112 (83.0%) patients of clear cell carcinoma, 1 (0.7%) patient of renal medullary carcinoma, 1 (0.7%) patient of MiT family translocation renal cell carcinoma and 15 patients of other types (including 6 patients with unclassified carcinoma, 6 patients with chromophobe carcinoma and 3 patients with other types).

In the univariate logistic regression analysis, age, ECOG score (to understand the general health status and indicators of treatment tolerance of patients from their physical strength, the range is 1–5 points), complaint, clinical symptoms, tumor morphology, tumor pseudocapsule, tumor necrosis, lymph node status from pre-operative imaging, lymph node size and fusion, distant metastasis, D-dimer, LDH, urine occult blood, urine protein, urinary tract infection and clinical T category were significantly correlated with a postoperative lymph node positive (Appendix A). Then we obtained nine variables according to lasso regression analysis (Figure 1, Table 2), including age, urine protein, lymph node status from pre-operative imaging, lymph node size category, pseudocapsule, urine occult blood, urinary tract fusion, lymph node fusion (on CT or MRI, it seems that lymph nodes adhere to each other and are multinodular) and clinical T stage (cT1–2 vs. cT3–4). Afterwards, multivariable logistic regression analysis was conducted and identified three independent predictors of lymph node metastasis (Table 2), among which age was a protective factor of lymph node metastasis (OR = 0.3769, 95% CI = 0.1864–0.7622, *p* < 0.01), and lymph node size category (10–20 mm: OR = 15.0040, 95% CI = 1.5666–143.7000, *p* <0.05; >20 mm: OR = 4.4013, 95% CI = 1.4892–7.3134, *p* < 0.01) and clinical T stage (cT1–2 vs. cT3–4) (OR = 3.1641, 95% CI = 1.0336–9.6860, *p* < 0.05) were two risk factors of lymph node metastasis.

Based on the above independent predictors, a nomogram was established to predict the probability of lymph node metastasis in patients with RCC before surgery. As shown in Figure 2, we randomly selected a patient and added the scores of the three independent factors. The total score was 6.9, and the corresponding probability of lymph node metastasis is 97.5%. The calibration curve showed there was no significant statistical difference between our prediction model and the ideal curve; that is, the model had good fitting (Figure 3a). Moreover, the AUC (Area Under Curve) of the ROC (Receiver Operating Characteristic Curve) for our model was 0.94, which was the largest among all the models (Figure 3b). Furthermore, the DCA of this model showed a threshold probability of 0–85%, in which our model and the single factor of lymph node size category can identify patients with possible lymph node metastasis, which is better than the “treat all patients” or “no treatment” schemes and other models based on univariable (Figure 4). Then we calculated the C-index of our model and the single factor of lymph node size category. The model was 0.94 (95% CI: 0.9–0.97), and the lymph node size was 0.88 (95% CI: 0.83–0.94), showing that the accuracy of the model’s prediction is greater than that of LN size prediction alone. In order to help the clinical decision, we drew a decision tree (Appendix A); see Appendix A for the production process of the tree.

## 4. Discussion

Lymph node infiltration is one of the most important predictors of a tumor’s progression and the patient’s death. The detection and treatment of lymph node metastasis is a key issue in clinical decision making. Imaging examination of hilar or retroperitoneal lymph nodes with a diameter of >2 cm is usually considered to have malignant characteristics, and lymph nodes with a diameter of >1 cm is usually considered to be suspicious of lymph node involvement. However, the existing imaging technology cannot reliably predict the status of lymph nodes of all sizes. The false positive rate is as high as 58%, and the false negative rate in predicting lymph node metastasis with a diameter of <1 cm is about 10% [11]. A previous retrospective study showed that among patients who underwent lymph node dissection during operation after suspicious lymph node metastasis or the discovery of enlarged lymph nodes found with a preoperative imaging examination, the positive rate of postoperative pathological lymph nodes was only 4–33% [6], and the rate of occult lymph node metastasis that could not be identified with an imaging examination was generally 3–13% [12]. In this study, the proportion of 189 patients with postoperative pN1 stage was 28.57% (54/189); the proportion of 97 patients with cN1 stage finally confirmed PN1 stage was 51.55% (50/97), and the proportion of lymph node metastasis not found with an imaging examination was 7.61% (7/92). These results were consistent with the conclusion of the previous retrospective studies.

Recent studies showed that whether in single-center or multi-center research, cytoreductive surgery can be carried out safely, can improve cachexia, can respond to systemic immunotherapy and may also improve survival [13]. In other words, for patients with localized RCC, the current view does not advocate extensive lymph node dissection. However, for patients with RCC at risk of lymph node metastasis, such as patients with enlarged lymph nodes, it is still controversial whether to perform lymph node dissection. On the one hand, blindly expanding the operation will increase the operation risk, damage the patient’s immunity and affect the postoperative quality of life. On the other hand, whether lymph node dissection can improve the survival time and quality of life of patients with clear lymph node metastasis is uncertain. Therefore, how to accurately identify patients who need lymph node dissection or biopsy is very important. The risk factors of lymph node metastasis have been widely studied globally. Capitanio et al. analyzed the data of 1983 patients with RCC and put forward a model to predict lymph node metastasis before operation. High clinical T stage, positive lymph nodes in imaging, distant metastasis and large tumor diameter were independent influencing factors of lymph node metastasis [14]. The results of Terrone et al. were similar [15]. Blute et al. [16] included 1652 patients with renal carcinoma who underwent radical nephrectomy, and multivariate analysis showed that the risk factors of lymph node metastasis of renal cancer were tumor cell grade 3–4, sarcomatous component, maximum tumor diameter > 10 cm, tumor stage pT3-pT4 and necrosis in tumor tissues. In addition, it was also found that patients with enlarged lymph nodes and advanced metastases who underwent cytoreductive surgery also had a higher rate of lymph node metastasis. In our study, high tumor T stage and large lymph node diameter in pre-operative imaging were independent risk factors of renal cell carcinoma with lymph node metastasis. After further dividing the lymph node size into three categories of <10 mm, 10–20 mm and >20 mm, we found that the risk of lymph node metastasis increased significantly with the increase of lymph node diameter. These results were consistent with other research results mentioned above. Moreover, Gershman et al. found that the risk of lymph node metastasis increased by 1.19 times for every 1 mm increase in the maximum lymph node diameter (95% CI = 1.13–1.25, *p* < 0.001) in a study evaluating the size of imaging lymph nodes to predict lymph node involvement [17]. Moreover, unlike results of other studies, in our study, age was negatively correlated with the risk of lymph node involvement, which was consistent with the conclusion reported by Li et al. [18]. This result may be related to different compositions of the included patients, such as different pathological subtypes. The cases with a positive lymph node in our study included 14.8% of MiT family translocated renal cell carcinoma, which is a subtype of RCC with low onset age and is prone to lymph node metastasis [19]. In addition, we speculated that it may also be related to lower differentiation and faster tumor growth. The specific mechanism needs to be further studied. According to the independent predictors of lymph node metastasis of RCC, we constructed a visual nomogram to predict the probability of lymph node metastasis of RCC. The C index and internal validation results showed that the model has good prediction performance.

In addition, we also found that microscopic hematuria was associated with lymph node metastasis in univariate analysis, but it was not an independent predictor of lymph node metastasis in multivariate analysis. Hematuria often indicates that the tumor invades the collecting system, which may indicate a higher clinical stage or a more invasive tumor, so the probability of lymph node involvement is higher. Gross hematuria is a subjective description, which is mostly intermittent and not objective enough, while microscopic hematuria may be more accurate and reliable.

Hemoglobin, platelet count, NLR, lymphocyte percentage, A/G, LDH, blood calcium, fibrinogen and other blood indicators are considered related to the prognosis or metastasis of renal cell carcinoma [20]. Another study also showed that local symptoms, high ECOG score, clinically suspected lymph node metastasis and elevated LDH were independent predictors of retroperitoneal lymph node metastasis in patients with RCC [21]. Although the results of univariate analysis in our study showed that the above indicators cannot be used as independent predictors, the value of these indicators in predicting lymph node metastasis needs to be further evaluated after more studies are included. In recent years, imaging omics has been applied more and more in urological carcinoma. It is reported that imaging omics combined with clinical features can improve the accuracy of predicting lymph node metastasis in bladder cancer and colorectal cancer, but it has not been applied to predict lymph node metastasis in patients with RCC [22]. In this study, univariate analysis showed that lymph node status in pre-operative imaging was associated with lymph node metastasis, which indicates that the combination of imaging omics and clinical features is expected to build a more accurate preoperative prediction model of lymph node metastasis of RCC in the future.

There are also some limitations in this study: the sample size was small; there might exist selection deviation, which may cause excessive accuracy results; and the indication and scope of lymph node dissection in patients depended partially on the surgeons’ preference and intraoperative decision making. Most of the included patients were considered to have a high risk of lymph node metastasis, which was not enough to represent all patients with RCC. Moreover, the pathological subtypes of renal cell carcinoma in the included population are inconsistent with the actual incidence rate of each subgroup, especially MIT. In terms of data analysis, this study only conducted internal validation and did not evaluate the reliability of the prediction model through external queue validation. Furthermore, there is no very clear standard for lymph node dissection, and the operations of different patients are performed by different doctors; therefore, lymph node dissection areas may be different, which may produce false negative lymph nodes and reduce the effectiveness of our prediction model.

In conclusion, the results of this paper are based on the preliminary findings of a relatively small cohort of patients, showing that low age, large lymph node size in pre-operative imaging and high clinical T stage were independent risk factors for lymph node metastasis of RCC. Low age is an unexpected result, while has also been reported in the past study. The factors leading to this result have been analyzed in the previous article, but more queue data are still needed to prove their effectiveness. The predictive nomogram model based on these three factors had good fitting, which would be helpful to predict the risk of lymph node metastasis of RCC and assist clinical decision making before operation. We suggest that when patients have the above three risk factors, lymph node dissection should be performed during the operation, and regular and close reexamination and follow-up should be carried out after the operation.

## 5. Conclusions

We identified three independent predictive factors including low age, large lymph node size in pre-operative imaging and high clinical T stage of pathological lymph node metastasis in patients with RCC. According to the data we collected at present, our predictive nomogram using these factors exhibited excellent discrimination and calibration. In the future, we will continue to collect data related to this study and constantly improve our prediction model.

## Figures and Tables

**Figure 1 jcm-12-00441-f001:**
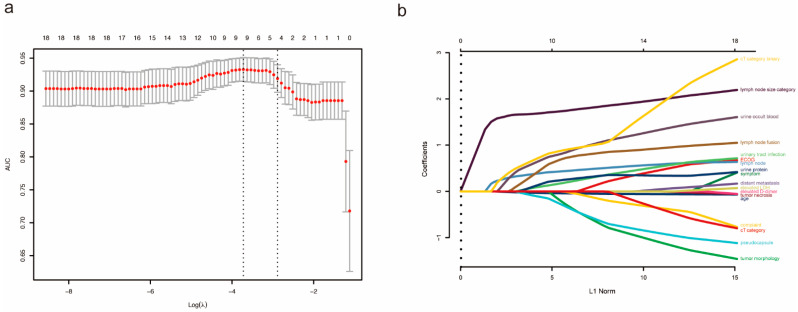
Result of lasso regression analysis. (**a**) Optimal predictor (lambda) selection in the lasso model. (**b**) lasso coefficient profiles of 18 predictors.

**Figure 2 jcm-12-00441-f002:**
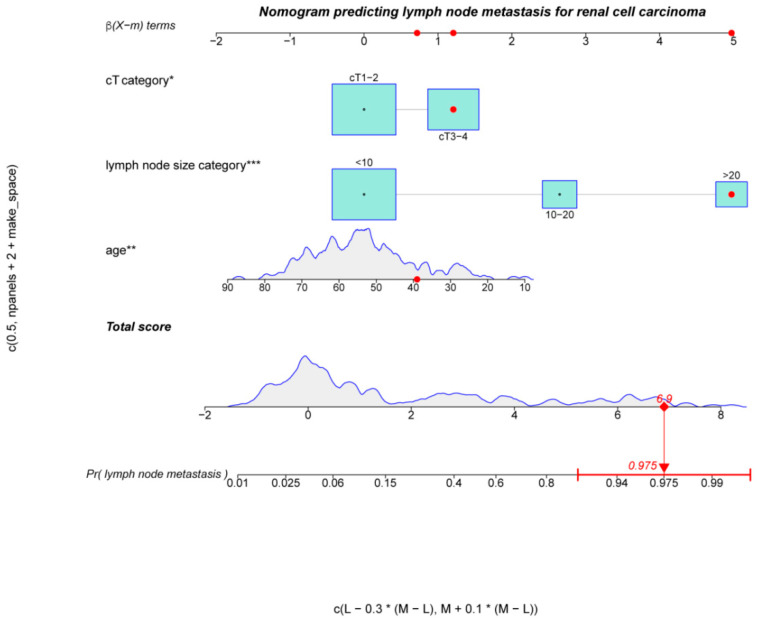
Nomogram of patients for predicting the risk of lymph node metastasis. Total score was 6.9, and the corresponding probability of lymph node metastasis was 97.5%. The asterisks represented the statistical *p* value (* *p* < 0.05; ** *p* < 0.01; *** *p* < 0.001).

**Figure 3 jcm-12-00441-f003:**
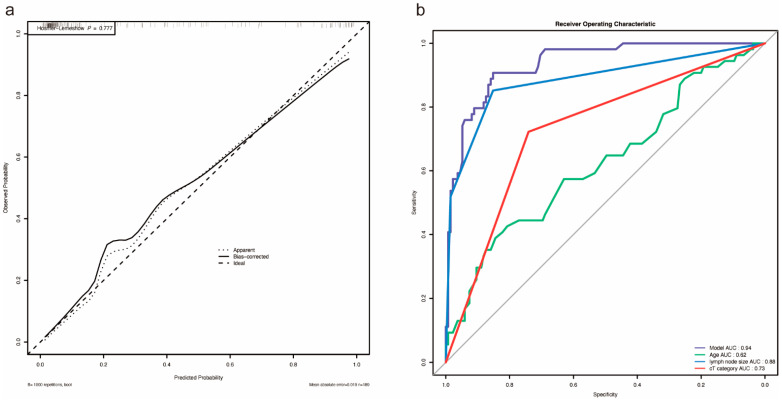
(**a**) Calibration curve for nomogram. (**b**) ROC curve. The area under AUC for the model was 0.94.

**Figure 4 jcm-12-00441-f004:**
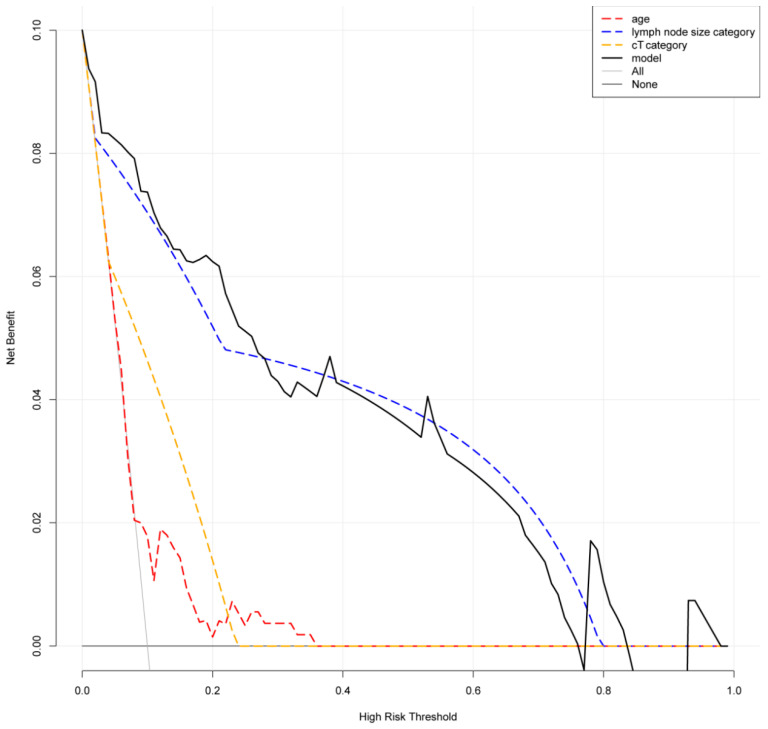
The DCA of this model showed a threshold probability of 0–85%.

**Table 1 jcm-12-00441-t001:** Basic characteristics of including patients.

	All Patients (n = 189)	pN0 (n = 135)	pN1 (n = 54)	*p* Value
**Gender, n (%)**				0.172
**Male**	121 (64.0)	91 (67.4)	30 (55.6)	
**Female**	68 (36.0)	44 (32.6)	24 (44.4)	
**Age, median [IQR] (years)**	54.00 [46.00, 63.00]	55.00 [48.00, 64.50]	51.00 [37.50, 61.00]	0.007
**ECOG PS, n (%)**				0.038 ^b^
**0**	131 (69.3)	100 (74.1)	31 (57.4)	
**1**	58 (30.7)	35 (25.9)	23 (42.6)	
**Complaint, n (%)**				0.006
**Medical examination**	92 (48.7)	75 (55.6)	17 (31.5)	
**Low back pain**	37 (19.6)	22 (16.3)	15 (27.8)	
**Hematuria**	36 (19.0)	26 (19.3)	10 (18.5)	
**Others**	24 (12.7)	12 (8.9)	12 (22.2)	
**Symptom, n (%)**				0.002
**None**	95 (50.3)	78 (57.8)	17 (31.5)	
**Local symptoms**	70 (37.0)	45 (33.3)	25 (46.3)	
**Systemic symptoms**	24 (12.7)	12 (8.9)	12 (22.2)	
**History of tumor, n (%)**				0.355 ^a^
**No**	183 (96.8)	132 (97.8)	51 (94.4)	
**Yes**	6 (3.2)	3 (2.2)	3 (5.6)	
**History of smoking and drinking, n (%)**				0.625
**No**	137 (72.5)	96 (71.1)	41 (75.9)	
**Yes**	52 (27.5)	39 (28.9)	13 (24.1)	
**History of abdominal surgery, n (%)**				1
**No**	162 (85.7)	116 (85.9)	46 (85.2)	
**Yes**	27 (14.3)	19 (14.1)	8 (14.8)	
**Hypertension, n (%)**				0.635
**No**	148 (78.3)	104 (77.0)	44 (81.5)	
**Yes**	41 (21.7)	31 (23.0)	10 (18.5)	
**Diabetes, n (%)**				1
**No**	165 (87.3)	118 (87.4)	47 (87.0)	
**Yes**	24 (12.7)	17 (12.6)	7 (13.0)	
**Coronary heart disease, n (%)**				1 ^a^
**No**	184 (97.4)	131 (97.0)	53 (98.1)	
**Yes**	5 (2.6)	4 (3.0)	1 (1.9)	
**Respiratory diseases, n (%)**				0.41 ^a^
**No**	182 (96.3)	131 (97.0)	51 (94.4)	
**Yes**	7 (3.7)	4 (3.0)	3 (5.6)	
**Infectious disease, n (%)**				1 ^a^
**No**	184 (97.4)	131 (97.0)	53 (98.1)	
**Yes**	5 (2.6)	4 (3.0)	1 (1.9)	
**Urolithiasis, n (%)**				0.45 ^a^
**No**	180 (95.2)	127 (94.1)	53 (98.1)	
**Yes**	9 (4.8)	8 (5.9)	1 (1.9)	
**aCCI, n (%)**				0.285 ^b^
**0–1**	102 (54.0)	68 (50.4)	34 (63.0)	
**2–3**	67 (35.4)	52 (38.5)	15 (27.8)	
**≥4**	20 (10.6)	15 (11.1)	5 (9.3)	
**Tumor side, n (%)**				0.121
**Left**	104 (55.0)	69 (51.1)	35 (64.8)	
**Right**	85 (45.0)	66 (48.9)	19 (35.2)	
**Tumor size, median [IQR] (mm)**	70.00 [50.00, 98.00]	69.00 [50.00, 92.00]	70.00 [46.00, 102.50]	0.739
**Exophytic or endophytic property, n (%)**				0.123 ^a^
**50% or more exophytic**	36 (19.0)	26 (19.3)	10 (18.5)	
**Less than 50% exophytic**	140 (74.1)	103 (76.3)	37 (68.5)	
**Entirely endophytic**	13 (6.9)	6 (4.4)	7 (13.0)	
**Renal collecting system under pressure, n (%)**				0.13
**No**	23 (12.2)	20 (14.8)	3 (5.6)	
**Yes**	166 (87.8)	115 (85.2)	51 (94.4)	
**Renal hilum invasion, n (%)**				0.103
**No**	64 (33.9)	51 (37.8)	13 (24.1)	
**Yes**	125 (66.1)	84 (62.2)	41 (75.9)	
**Tumor morphology, n (%)**				<0.001
**Round**	74 (39.2)	65 (48.1)	9 (16.7)	
**Irregular**	115 (60.8)	70 (51.9)	45 (83.3)	
**Pseudocapsule, n (%)**				<0.001
**No**	87 (46.0)	45 (33.3)	42 (77.8)	
**Yes**	102 (54.0)	90 (66.7)	12 (22.2)	
**Tumor necrosis, n (%)**				0.062
**No**	35 (18.5)	30 (22.2)	5 (9.3)	
**Yes**	154 (81.5)	105 (77.8)	49 (90.7)	
**Venous tumor thrombus, n (%)**				0.464 ^a^
**No**	162 (85.7)	118 (87.4)	44 (81.5)	
**Renal vein**	26 (13.8)	16 (11.9)	10 (18.5)	
**Vena cava**	1 (0.5)	1 (0.7)	0 (0.0)	
**Lymph node status by pre-operative imaging, n (%)**				<0.001
**Negative**	92 (48.7)	85 (63.0)	7 (13.0)	
**Enlarged and increased lymph nodes**	56 (29.6)	44 (32.6)	12 (22.2)	
**Metastasis**	41 (21.7)	6 (4.4)	35 (64.8)	
**Lymph node size by pre-operative imaging, median [IQR] (mm)**	5.00 [0.00, 15.00]	0.00 [0.00, 8.00]	21.00 [17.00, 30.00]	<0.001
**Lymph node size category, n (%)**				<0.001 ^b^
**≤10 mm**	123 (65.1)	115 (85.2)	8 (14.8)	
**10–20 mm**	36 (19.0)	18 (13.3)	18 (33.3)	
**>20 mm**	30 (15.9)	2 (1.5)	28 (51.9)	
**Lymph node fusion, n (%)**				<0.001
**No**	170 (89.9)	134 (99.3)	36 (66.7)	
**Yes**	19 (10.1)	1 (0.7)	18 (33.3)	
**Clinical M stage, n (%)**				0.005 ^b^
**cM0**	163 (86.2)	123 (91.1)	40 (74.1)	
**cM1**	26 (13.8)	12 (8.9)	14 (25.9)	
**Neutrophil, median [IQR] (10^9^ cells/L)**	3.97 [3.10, 5.11]	3.87 [2.90, 5.08]	4.33 [3.44, 5.12]	0.128
**Neutrophil, n (%)**				1
**Normal**	166 (87.8)	119 (88.1)	47 (87.0)	
**Abnormal**	23 (12.2)	16 (11.9)	7 (13.0)	
**Lymphocyte, mean (SD) (10^9^ cells/L)**	1.60 (0.50)	1.62 (0.52)	1.56 (0.45)	0.523
**Lymphocyte, n (%)**				0.975
**Normal**	159 (84.1)	113 (83.7)	46 (85.2)	
**Abnormal**	30 (15.9)	22 (16.3)	8 (14.8)	
**NLR, median [IQR]**	2.57 [1.82, 3.68]	2.51 [1.73, 3.50]	2.73 [2.04, 4.00]	0.181
**Hemoglobin, median [IQR] (g/L)**	126.00 [104.00, 139.00]	127.00 [108.00, 140.50]	120.00 [102.00, 136.00]	0.206
**Anemia, n (%)**				0.447
**No**	101 (53.4)	75 (55.6)	26 (48.1)	
**Yes**	88 (46.6)	60 (44.4)	28 (51.9)	
**Platelet, median [IQR] (10^9^ cells/L)**	258.00 [190.00, 315.00]	248.00 [186.00, 310.50]	272.00 [195.00, 330.50]	0.136
**Platelet, n (%)**				0.747
**Normal**	153 (81.0)	108 (80.0)	45 (83.3)	
**Abnormal**	36 (19.0)	27 (20.0)	9 (16.7)	
**Fibrinogen, median [IQR] (g/L)**	4.31 [3.22, 6.13]	4.08 [3.09, 6.06]	4.96 [3.60, 6.22]	0.139
**Fibrinogen, n (%)**				0.156
**Normal**	80 (42.3)	62 (45.9)	18 (33.3)	
**Abnormal**	109 (57.7)	73 (54.1)	36 (66.7)	
**D-dimer, median [IQR] (mg/L)**	0.55 [0.30, 0.97]	0.46 [0.30, 0.92]	0.66 [0.38, 1.09]	0.073
**D-dimer, n (%)**				0.011
**Normal**	89 (47.1)	72 (53.3)	17 (31.5)	
**High**	100 (52.9)	63 (46.7)	37 (68.5)	
**Clotting time, n (%)**				0.568
**Normal**	130 (68.8)	95 (70.4)	35 (64.8)	
**Prolonged**	59 (31.2)	40 (29.6)	19 (35.2)	
**Albumin, median [IQR] (g/L)**	39.00 [35.10, 41.40]	39.40 [35.80, 41.80]	38.35 [34.73, 40.30]	0.096
**Albumin, n (%)**				0.641
**Normal**	146 (77.2)	106 (78.5)	40 (74.1)	
**Low**	43 (22.8)	29 (21.5)	14 (25.9)	
**Globulin, median [IQR] (g/L)**	31.80 [27.60, 37.00]	30.90 [27.25, 36.00]	33.15 [28.75, 39.65]	0.173
**Globulin, n (%)**				0.132
**Normal**	129 (68.3)	97 (71.9)	32 (59.3)	
**High**	60 (31.7)	38 (28.1)	22 (40.7)	
**AGR, mean (SD)**	1.23 (0.36)	1.26 (0.36)	1.17 (0.36)	0.155
**AGR < 1.5, n (%)**				0.321
**No**	46 (24.3)	36 (26.7)	10 (18.5)	
**Yes**	143 (75.7)	99 (73.3)	44 (81.5)	
**ALP, median [IQR] (U/L)**	72.00 [61.00, 94.00]	71.00 [61.00, 91.00]	77.50 [63.00, 98.00]	0.49
**ALP, n (%)**				0.239 ^a^
**Normal**	174 (92.1)	122 (90.4)	52 (96.3)	
**High**	15 (7.9)	13 (9.6)	2 (3.7)	
**LDH, median [IQR] (U/L)**	171.00 [143.00, 200.00]	167.00 [138.00, 189.50]	178.00 [156.75, 253.75]	0.006
**LDH, n (%)**				0.004
**Normal**	155 (82.0)	118 (87.4)	37 (68.5)	
**High**	34 (18.0)	17 (12.6)	17 (31.5)	
**Calcium, median [IQR] (mmol/L)**	2.38 [2.30, 2.49]	2.37 [2.29, 2.48]	2.43 [2.36, 2.52]	0.021
**Calcium, n (%)**				0.427
**Normal**	165 (87.3)	120 (88.9)	45 (83.3)	
**High**	24 (12.7)	15 (11.1)	9 (16.7)	
**Creatinine, median [IQR] (μmol/L)**	76.00 [66.00, 92.00]	76.00 [66.00, 93.00]	79.00 [65.00, 90.75]	0.962
**Creatine, n (%)**				1
**Normal**	159 (84.1)	114 (84.4)	45 (83.3)	
**High**	30 (15.9)	21 (15.6)	9 (16.7)	
**eGFR, median [IQR]**	91.70 [73.80, 102.90]	92.60 [73.80, 100.65]	88.75 [74.02, 104.80]	0.753
**Urine RBC, median [IQR] (cells/hpf)**	10.20 [3.00, 49.90]	8.10 [3.00, 30.00]	31.50 [2.88, 416.50]	0.012
**Urine occult blood, n (%)**				0.001
**No**	117 (61.9)	94 (69.6)	23 (42.6)	
**Yes**	72 (38.1)	41 (30.4)	31 (57.4)	
**Urine WBC, median [IQR] (cells/hpf)**	6.30 [1.70, 35.50]	6.00 [1.70, 19.85]	11.50 [1.35, 97.30]	0.075
**Urinary tract infection, n (%)**				0.015
**No**	128 (67.7)	99 (73.3)	29 (53.7)	
**Yes**	61 (32.3)	36 (26.7)	25 (46.3)	
**Urine protein, n (%)**				0.001
**No**	145 (76.7)	113 (83.7)	32 (59.3)	
**Yes**	44 (23.3)	22 (16.3)	22 (40.7)	
**Surgery type and technique, n (%)**				<0.001 ^a^
**Open radical nephrectomy**	67 (35.4)	37 (27.4)	30 (55.6)	
**Open partial nephrectomy**	2 (1.1)	2 (1.5)	0 (0.0)	
**Laparoscopic radical nephrectomy**	88 (46.6)	75 (55.6)	13 (24.1)	
**Laparoscopic partial nephrectomy**	10 (5.3)	10 (7.4)	0 (0.0)	
**Robot-assisted radical nephrectomy**	17 (9.0)	7 (5.2)	10 (18.5)	
**Robot-assisted partial nephrectomy**	5 (2.6)	4 (3.0)	1 (1.9)	
**Resected lymph nodes, median [IQR]**	2.00 [1.00, 6.00]	2.00 [1.00, 6.00]	3.00 [1.00, 8.75]	0.162
**Positive lymph nodes, median [IQR]**	0.00 [0.00, 1.00]	0.00 [0.00, 0.00]	2.00 [1.00, 3.00]	<0.001
**Clinical T stage, n (%)**				<0.001 ^b^
**cT1**	75 (39.7)	64 (47.4)	11 (20.4)	
**cT2**	40 (21.2)	36 (26.7)	4 (7.4)	
**cT3**	56 (29.6)	28 (20.7)	28 (51.9)	
**cT4**	18 (9.5)	7 (5.2)	11 (20.4)	
**Histology type, n (%)**				<0.001 ^a^
**Clear cell carcinoma**	126 (66.7)	112 (83.0)	14 (25.9)	
**Medullary carcinoma**	2 (1.1)	1 (0.7)	1 (1.9)	
**Papillary**	25 (13.2)	6 (4.4)	19 (35.2)	
**MiT**	9 (4.8)	1 (0.7)	8 (14.8)	
**Collecting duct carcinoma**	3 (1.6)	0 (0.0)	3 (5.6)	
**Unclassified**	15 (7.9)	6 (4.4)	9 (16.7)	
**Chromophobe**	6 (3.2)	6 (4.4)	0 (0.0)	
**Others**	3 (1.6)	3 (2.2)	0 (0.0)	

IQR, interquartile range; SD, standard deviation, ECOG PS, Eastern Cooperative Oncology Group performance status; aCCI, age-adjusted Charlson Comorbidity Index; NLR, neutrophil lymphocyte ratio; AGR, albumin globulin ratio; ALP, alkaline phosphatase; LDH, lactate dehydrogenase; RBC, red blood cell; WBC, white blood cell. ^a^ Tested with Fisher’s Exact Test. ^b^ Tested with Kruskal-Wallis H Test.

**Table 2 jcm-12-00441-t002:** Multivariable logistic regression analysis of predictors of lymph node metastasis.

Variables	B	SE	OR	95% CI	*p* Value
**Age**	−0.0574	0.0211	0.3769	[0.1864–0.7622]	0.0066
**Urine protein**	0.6814	0.6861	1.9766	[0.5151–7.5841]	0.3207
**Lymph node status in pre-operative imaging**					
**Negative**	-	-	-	-	Ref.
**Enlarged and increased**	−1.5404	1.1512	0.2143	[0.0224–2.0460]	0.1809
**Metastasis**	0.4349	1.3311	1.5449	[0.1137–20.9840]	0.7438
**Lymph node size category**					
≤10	-	-	-	-	Ref.
10–20	2.7083	1.1528	15.0040	[1.5666–143.7000]	0.0188
>20	4.4013	1.4858	4.4013	[1.4892–7.3134]	0.0031
**Pseudocapsule (Yes vs. No)**	−0.4616	0.5898	1.5866	[0.4994–5.0410]	0.4338
**Urine occult blood (Positive vs. Negative)**	1.3974	0.7238	4.0445	[0.9790–16.7090]	0.0535
**Urinary tract infection (Yes vs. No)**	0.2208	0.6332	1.2471	[0.3605–4.3141]	0.7273
**Lymph node fusion (Yes vs. No)**	0.7686	1.3527	2.1566	[0.1522–30.5660]	0.5699
**Clinical T stage (cT1–2 vs. cT3–4)**	1.1519	0.5708	3.1641	[1.0336–9.6860]	0.0436

B, regression coefficient; SE, standard error; OR, Odds Risk; CI, confidence interval.

## Data Availability

All the original data were collected from Tongji Hospital, Tongji Medical College, Huazhong University of Science and Technology.

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
