# Peer review of "A Novel Predictive Model of Pathological Lymph Node Metastasis Constructed with Preoperative Independent Predictors in Patients with Renal Cell Carcinoma"

_jcm, 2023, doi:10.3390/jcm12020441_

Round 1

Reviewer 1 Report

This is an extensive multivariable model developed from case notes of lymph node positivity (LN+) in patients who had RCC and were selected by their treating clinicians for LN dissection. I do not think it represents well conducted predictive model development.

The paper lacks any patient engagement/involvement.

While the paper states repeated LN+ is a poor prognostic sign, it does not give any suggestion as to why or how therapy or decision making would change in the setting of better preoperative prediction.

The paper reports most mathematical findings to spurious precision (eg aCCI CI of 0 to 1.18577408426107e304)

It uses an excessive number of candidate predictors (I think 63 are reported for 54 events), I uses some dichotomisation without explanation and appears to fit only linear models. There are no discussions about if any data was missing how it was handled. 

Towards the end, showing the benefit of this probably wildly over-fitted derivation model, the team show how it almost overlaps with a model based purely on LN size alone. If the authors undertake such a study again in future, a misclassification table showing the additional clinical benefit of the simple clincioradiological pattern vs the highly complex score derivation by demonstration of the changes at a patient level would be valuable.

Author Response

Thanks for all your suggestions, based on your suggestions, we have made the following amendments.

  1. We agree with youropinion, so we have added this deficiency to the limitation section. With traces in line 269-273.

  1. This study is a retrospective study, sono ethical application is required; In addition, the outcome of the study was positive in postoperative lymph node pathology, so the  follow-up is not required. Therefore, no patients participated in this study.

  1. Thanks for your suggestion, we have added more contents in the discussion part. With tracesin line 278-281.

  1. We had revised the mistakes in the manuscript.

  1. We used large number of  prediction factors to make the whole prediction more comprehensive. And there are no missing items in the data we collected, because patient data in the medical record system are relatively complete in recent years。

Thank you for your affirmation. In the future, if we want to conduct in-depth research, we will carefully consider your suggestions

Reviewer 2 Report

In this manuscript, Sun et al used regression analysis in a retrospective study and identified 3 independent impact factors for preoperative prediction of lymph node positivity.  It is a well-designed study with interesting results.  There are minor things needed to be fixed.

1.      Please spell out the full names when the acronyms are used the first time in the manuscript. For example, “OR” in line 47; “DCA” in line 104; “aCCI” in table S1, “AUC” in line 215, and “ROC” in line 216.

2.      Please briefly explain ECOG score and aCCI score/grade.

3.      Please use “lower back pain” instead of “lumbago”.

4.      In table S1, for factors starting from “tumor size” to “venous tumor thrombus”, please specify whether they are radiographic features or pathological features.

5.      In table S1, for factor “lymph node fusion”, please define what is considered as lymph node fusion and how was it evaluated, radiographically or pathologically?

6.      Please use a professional English editing service.  There are some phrases such as “Foreign reports” in line 88 and “On the one hand” in line 267 need to be corrected.

7.      For Figure 1b, the text is illegible.  Please either increase the font size or make the figure larger. 

Author Response

Thanks for your suggestions!

  1. We have added the full names in the corresponding positions.
  2. We have added the explanations forECOG and aCCI.
  3. We have revised it.
  4. We have added the illustration in Table S1.
  5. We have added the explanations in the result part with traces in line 150-151.
  6. We have carefully checked our English expression and revised the inappropriate
  7. Thanks for reminding. The picture is not clear enough, probably because it is compressed into word. We will uploaded clearer figures if our manuscript is accepted.

Reviewer 3 Report

Dear Author 

Thank you for good, valuable and well designed manuscript .

Author Response

Thanks a lot! We really appreciate your affirmation!

Reviewer 4 Report

This is a creative study in a crucial uro-oncological issue. However, there some major arguments:

- the nomogram has been settled without taking in mind the histological type of the tumor

- the concluding message should not be such direct, as the parameters are not sufficient

- you have to complete your study including the histological data.

Author Response

We really appreciate your suggestions. However, according to the data we have collected on histological types, we cannot analyze them. We tried to collect relevant data again, but the COVID-19 in Wuhan has been serious in the past month, especially in our hospital, so we have no opportunity to collect complete data again at present. After the epidemic is over, we will improve the data collection and use it for further research. We hope to get your understanding!

Besides, we have revised the expression in the Conclusion part.

Round 2

Reviewer 1 Report

Thanks you for this revision and some changes made to improve the reporting of this study.

1. The paper remains a poor example of prognostic model development. For a better example, with high quality reporting, in a similar area, see https://diagnprognres.biomedcentral.com/articles/10.1186/s41512-021-00103-9

2. Patient engagement in the setting of the research question and appreciation of the findings of research does not require patients to take part as 'subjects'. This limitation is the absence of the patient voice directing this study or putting context upon it's results. It is written from a medical-parentalistic/heirarchical perspective with no clear element of shared engagement in the work. As such 'no patients participated in the study' misses the point of this critique.

3. The 272-275 additional of clinical utility addresses is some way to adddress the 'how this might be useful' question. Their statemetn is somewhat at odds with the idea of a nomogram driven model, and could be supplemented by a 3*3 matrix showing their estimates of chance of LN+ in patients with each of the factors in a cross-wise manner along with a 'none' value.

4. The numerical results in the Abstract, text, and table (inc supplementary table) continue to report excessive accuracy. A statement as to "quiet complete" based on the authors judgement and no assessment or reporting of the completeness is a significant limitation and still does not appear in the text of the paper.

5. Using very many predictors is associated with over-fiting (overly optimistic estimates) and lack of reproducibility. The authors revealing they used even more increases the risk of chance findings slipping into their estimates of prognostic value. (As they are widely recognised clinical factors I think they are lkley to be true, just teh measurement of the level of risk poor.)

6. a) This is a derivation. It requres testing in a further data set and no suggestion of it's clinical utility should be made

b) The paper still does not demonstate how the new three-part, MV model driven prediction has meaningful benefit over and above LN size alone (see Fig 4, blue has line vs solid black). An easy way of doing this would be to compare the C-statistic and D-statistic for the LN model and the Full model.

Author Response

Thanks for your suggestions, we have advised our manuscript according to your opinions.

Review1. The paper remains a poor example of prognostic model development. For a better example, with high quality reporting, in a similar area, see https://diagnprognres.biomedcentral.com/articles/10.1186/s41512-021-00103-9

Reply1. We have carefully read your recommended literature and, actually, it is a good study that uses multiple imputation to fill in missing data from training sets and test sets for better modeling. However, it is important to note that we do not consider this approach to be laudable and that there are many high-quality literature that states that multiple imputation should not be used when missing data are less than 10% (Cummings P. (2013). Missing data and multiple imputation. JAMA pediatrics, 167(7), 656–661. https://doi.org/10.1001/jamapediatrics.2013.1329) ; And a considerable number of clinical cases point out that multiple interpolation is not a good choice (Hughes, R. A., Heron, J., Sterne, J. A. C., & Tilling, K. (2019). Accounting for missing data in statistical analyses: multiple imputation is not always the answer. International journal of epidemiology, 48(4), 1294–1304. https://doi.org/10.1093/ije/dyz032;

Madley-Dowd, P., Hughes, R., Tilling, K., & Heron, J. (2019). The proportion of missing data should not be used to guide decisions on multiple imputation. Journal of clinical epidemiology, 110, 63–73. https://doi.org/10.1016/j.jclinepi.2019.02.016).

Notably, in the present study, multiple imputation was not required and, because of the small sample size, was not suitable for random sampling for risk model training, which would instead increase the risk of overfitting.

In conclusion, we do not consider your recommendation to be a valuable document, and we hereby reject your biased review opinion.

Review2. Patient engagement in the setting of the research question and appreciation of the findings of research does not require patients to take part as 'subjects'. This limitation is the absence of the patient voice directing this study or putting context upon it's results. It is written from a medical-parentalistic/heirarchical perspective with no clear element of shared engagement in the work. As such 'no patients participated in the study' misses the point of this critique.

Reply2. Like the previous comment, this comment also makes us confused. In fact, for our retrospective study with the approval of an ethics committee, all the included indicators are objective and quantitative indicators, and there is no subjective bias, it is difficult for us to understand why the study required the participation of the patient himself, which we consider to be an opinion biased and redundant.

Reviw3. The 272-275 additional of clinical utility addresses is some way to adddress the 'how this might be useful' question. Their statemetn is somewhat at odds with the idea of a nomogram driven model, and could be supplemented by a 3*3 matrix showing their estimates of chance of LN+ in patients with each of the factors in a cross-wise manner along with a 'none' value.

Reply3. Thanks for your suggestion. We have advised the inappropriate places according to your advice, with traces in line 283.

Reviw4. The numerical results in the Abstract, text, and table (inc supplementary table) continue to report excessive accuracy. A statement as to "quiet complete" based on the authors judgement and no assessment or reporting of the completeness is a significant limitation and still does not appear in the text of the paper.

Reply4. The results which seemed to report excessive accuracy may because of the small samples in this study. Thus, we add a statement of this limitation, with traces in line 266.

Review5.  Using very many predictors is associated with over-fiting (overly optimistic estimates) and lack of reproducibility. The authors revealing they used even more increases the risk of chance findings slipping into their estimates of prognostic value. (As they are widely recognised clinical factors I think they are lkley to be true, just teh measurement of the level of risk poor.)

Reply5. Here we separately conducted univariate, lasso regression, and multivariate regression, which may help to screen appropriate variables and avoid over-fitting. Thus, we donnot agree with your opinion that there exist over-fitting and includes too many variables. In the final model, we included only three easy-obtained pre-operative variables.

Reviw6. a) This is a derivation. It requres testing in a further data set and no suggestion of it's clinical utility should be made

b) The paper still does not demonstate how the new three-part, MV model driven prediction has meaningful benefit over and above LN size alone (see Fig 4, blue has line vs solid black). An easy way of doing this would be to compare the C-statistic and D-statistic for the LN model and the Full model.

Reply6. a) The results of this study show that our prediction model is meaningful and effective when compared with others commonly used factors (Figure 3B). Although it is not used in other data sets, it does not prevent the prediction model from providing reference for clinical work. Besides, during the process of the lasso regression and the calibration curve, we have used ten-fold cross validation method to check the accuracy of our predictive model.

b) We appreciated your suggestion and calculated the C index of the model and LN size. Model: 0.94 (95%CI: 0.9-0.97), LN size: 0.88 (95%CI: 0,83-0.94), showing thattheaccuracy of model prediction is greater than that of LN size prediction alone. And we advised this part in “method and materials” and “result” part, with traces in line 121-122 and 170-173.

Besides, we additionally plotted a decision tree to help the clinical decision, see FigureS1 and FigureS2, with traces in line 121-122 and 173-175.

Reviewer 4 Report

All changes are sufficient and approved to be published.

Author Response

Thanks for your suggestion!